# Convenient Decision Criteria for Surgery in Elderly Patients with Oral Squamous Cell Carcinoma

**DOI:** 10.3390/dj11010006

**Published:** 2022-12-26

**Authors:** Ryo Takasaki, Kenji Yamagata, Satoshi Fukuzawa, Fumihiko Uchida, Naomi Ishibashi-Kanno, Hiroki Bukawa

**Affiliations:** Department of Oral and Maxillofacial Surgery, Institute of Clinical Medicine, Faculty of Medicine, University of Tsukuba, 1-1-1 Tennodai, Tsukuba 305-8575, Japan

**Keywords:** oral squamous cell carcinoma (OSCC), elderly patients, surgery, decision tree analysis, performance status (PS), serum albumin (Alb), clinical stage

## Abstract

Elderly patients with oral squamous cell carcinoma (OSCC) have difficulty undergoing curative surgical treatment due to various factors besides age. The purpose of the present study was to study the factors determining surgery in elderly patients with OSCC. We designed and implemented a retrospective cohort study. The study sample included elderly patients aged ≥ 70 years with OSCC and they were statistically compared between the surgery and non-surgery groups. The primary outcome variable was selecting surgery as the treatment plan, while the secondary outcome was the prognosis of each group. The sample comprised 76 patients aged ≥ 70 years with OSCC, of whom 52 treated with surgery and 24 patients treated with non-surgery. As decision factors, performance status (PS), clinical stage, serum Alb level, body mass index (BMI), and Geriatric Nutritional Risk Index (GNRI) were significantly associated with the selection of surgery. Logistic multivariate analysis identified three independent predictive factors for selecting surgery: Alb (≥3.5 vs. <3.5), PS (0, 1, 2, 3), and clinical stage. According to the decision tree analysis, curative surgery is the recommended treatment strategy for elderly patients with Alb ≥ 3.5 g/dL, PS 0, and stage I, II. In conclusion, Alb, PS, and clinical stage may be the criteria for selecting surgery in elderly patients.

## 1. Introduction

Worldwide, approximately 380,000 new cases of oral squamous cell carcinoma (OSCC) were diagnosed by 2020 [1]. As the population ages, the number of elderly patients with OSCC is expected to further increase. The treatment strategy for OSCC is determined by the disease progression, age, and the patient’s physical status, nutritional status, and desire [2]. Surgery is the optimal treatment for patients with resectable OSCC and anticipated good prognosis [3]. Although the treatment outcomes with surgical therapy in elderly OSCC patients are nearly the same as those in younger patients, it is unclear whether curative surgery, which may contribute to risks and toxicities, is beneficial for elderly patients [4]. The chronologic age by itself is thought to be an unreliable parameter for decision-making [5].

Grénman et al. reported that several studies have shown that elderly patients with head and neck cancer (HANC) are less likely to receive potentially curative treatment than the younger age group based on age alone. However, the Panel concluded that chronologic age by itself is an unreliable parameter for decision-making. The physical status and psychological profile are more important factors, and a thorough preoperative evaluation and risk assessment are warranted. Therefore, elderly patients should be offered treatment options with curative intent like younger patients [6].

In this study, we retrospectively reviewed the treatment strategies and outcomes to investigate factors that significantly impact decision-making about the treatment strategies using the decision tree analysis and resultant treatment outcomes among elderly patients ≥ 70 years. This study aimed to determine significant factors that select the curative surgery or non-surgical treatment for proper overall survival (OS) in elderly patients.

## 2. Materials and Methods

This retrospective cohort study included patients aged 70 years or older and diagnosed with OSCC who visited the Department of Oral and Maxillofacial Surgery, the University of Tsukuba Hospital during the 4 years from 2015 to 2018.

The TN classification and stage classification were based on the 8th edition of the TNM classification of the UICC. Performance status (PS) was determined according to the Eastern Cooperative Oncology Group (ECOG) score during medical interview by oral surgeons as follows: PS 0: fully active, able to carry on all pre-disease performance without restriction. PS 1: restricted in physically strenuous activity but ambulatory and able to carry out work of a light or sedentary nature. PS 2: ambulatory and capable of all selfcare but unable to carry out any work activities; up and about more than 50% of waking hours. PS 3: capable of only limited selfcare; confined to bed or chair more than 50% of waking hours. PS 4: completely disabled; cannot carry on any selfcare; totally confined to bed or chair [7]. The primary outcome variable was selecting surgery as the treatment plan, while the secondary outcome was the prognosis of each group. The primary predictor variables were serum albumin, PS, and clinical stage.

Surgery is preferred as a curative treatment for patients with OSCC. Radical surgery with or without chemo/radiotherapy was performed according to the patient’s condition. Chemotherapy was administered to patients < 80 years old with a good general condition. Non-surgical treatment was radiotherapy as the almost noncurative one or best supportive care. The treatment was selected based on tumor stage, medical condition, PS, activities of daily living, and patient’s choice.

The following data were statistically compared between the surgery and non-surgery groups: sex, age, PS, presence of underlying disease, primary site, TN classification, stage classification, nutritional status [serum albumin (Alb) level, Geriatric Nutritional Risk Index (GNRI) [8], body mass index (BMI)], Charlson Comorbidity Index (CCI) updated version [9], and survival rate.

Chi-square tests were used for comparisons of sex, age, PS, present illness, primary site, TN classification, and nutritional status for the surgery and non-surgery groups. Survival curves were generated using the Kaplan–Meier method, and the Log-rank test was used for testing. Logistic multivariate analysis of the parameters with a stepwise forward selection method for the surgery or non-surgery group was used to identify independent variables for decision tree analysis. For investigating the clinical significance of any clinical factors on treatment decisions of surgery or non-surgery, decision tree analyses were performed with a selected variable of stepwise forward selection. All statistical analysis was performed with SPSS software ver. 28 (IBM Corp., Armonk, NY, USA). The criterion for determining statistically significant differences was *p* < 0.05.

This study was conducted as per the Declaration of Helsinki and was approved by the Institutional Review Board of the University of Tsukuba Hospital. Informed consent was waived due to the retrospective nature of the study (No. H29-258).

## 3. Results

Of 126 patients with primary OSCC that visited the Department of Oral and Maxillofacial Surgery, the University of Tsukuba Hospital during the 4 years from 2015 to 2018, 76 patients (45 males and 31 females) aged 70 years or older were included in this study. Of these patients, 52 (68.4%) were treated with surgery for curative treatment, and 24 (9 with only radiotherapy, 7 with chemoradiotherapy and 8 with palliative treatment) were not treated with surgery.

In the surgery group, 34 (63.1%) were males, and 18 (36.9%) were females; in the non-surgery group, 11 (46.9%) were males, and 13 (53.1%) were females. The median age at the first visit was 79.0 years (70–95 years). Patients were divided into four groups according to their age: 70 to 74 years old (24 cases), 75 to 79 years old (16 cases), 80 to 84 years old (17 cases), and 85 years old or older (19 cases). There was no significant difference in selected treatment between the four groups (Table 1). Regarding the primary site, the mandibular gingiva was the most common (21 patients, 40.4%), followed by the tongue (17 patients, 32.7%) in the surgery group. In the non-surgery group, mandibular gingiva was also the most common (10 patients, 41.7%), followed by the buccal mucosa (5 patients, 20.8%). There was no significant difference in treatment between the two groups.

The results for each category are shown in Table 1. There was a significant difference in PS between the surgery and non-surgery groups (*p* = 0.004). In the surgery group, PS 0 was present in 47 patients (90.4%), PS 1 in 4 (7.7%), and PS 3 in 1 (1.9%). On the other hand, in the non-surgery group, PS 0 was present in 14 patients (58.3%), PS 1 in 5 (20.8%), PS 2 in 4 (16.7%), and PS 3 in 1 (4.2%). PS 0 was the most common in both groups. There were significant differences in each TN and stage classification between both groups. Regarding T classification, the surgery group had the highest number of T2 cases in 18 (34.6%), followed by each of T1 and T4a in 14 cases (26.9%). In the non-surgery group, T4a was the most common with 11 cases (45.8%), followed by T4b with 6 cases (25.0%). Significant differences were observed in T classification between the surgery and non-surgery groups (*p* = 0.002). Regarding N classification, 36 patients (69.2%) in the surgery group had N0, 10 (19.2%) had N2b, and 6 (11.5%) had N1. In the non-surgery group, 10 (41.7%) had N0, 6 (25.0%) had N2b, and 5 (20.8%) had N2c. There were significant differences in N classification between both groups (*p* = 0.004). In terms of stage classification, Stage IVA was the most common in the surgery group with 18 cases (34.6%), followed by Stage II with 16 cases (30.8%) and Stage I with 14 cases (26.9%). In the non-surgery group, 11 (45.8%) of the patients were in Stage IVA, 6 (25.0%) were in Stage IVB, and 4 (16.7%) were in Stage III. A significant difference in clinical stages was observed between the surgery and non-surgery groups (*p* < 0.001).

The patients were classified thus: ≥ 3.5 g/dL as normal nutritional state and < 3.5 g/dL as undernourished in serum Alb level according to previous report [8]. In the surgery group, 3 patients (5.8%) had < 3.5 g/dL and 49 (94.2%) had ≥ 3.5 g/dL, while 10 (41.7%) had < 3.5 g/dL and 14 (58.3%) had ≥ 3.5 g/dL in the non-surgery group. A significant difference was observed in serum Alb levels between both groups (*p* < 0.001). Moreover, the patients were classified as underweight (BMI < 18.5 kg/m^2^), normal (BMI 18.5 to 25 kg/m^2^), and obese (BMI ≥ 25). In the surgery group, 5 (9.6%) were underweight, 28 (53.8%) had normal weight, and 19 (36.5%) were obese. In the non-surgery group, 3 (12.5%) were underweight, 14 (58.3%) had normal weight, and 1 (4.2%) was obese. There was a significant difference in BMI between both groups (*p* = 0.042).

For GNRI, a GNRI < 98 was considered at risk for nutritional disorders, and a GNRI ≥ 98 was for no nutritional disorders. In the surgery group, 6 (11.5%) had a GNRI < 98, and 46 (88.5%) had a GNRI ≥ 98, while in the non-surgery group, 7 (38.9%) had a GNRI < 98, and 11 (61.1%) had a GNRI ≥ 98. There was a significant difference in GNRI between both groups (*p* = 0.010).

The 5-year OS rate for the age groups from 70–74 years old was 79.6%, 75–79 years old was 61.1%, 80–84 years old was 50.5%, and over 85 years old was 27.1%. There was no significant difference between the age groups (Figure 1).

In contrast, 5-year OS rates for each treatment modality were 82.7% for surgery, 0% for radiotherapy, and 18.8% for palliative therapy (*p* < 0.01; Figure 2). Logistic multivariate analysis of the parameters with stepwise forward selection method identified three independent predictive factors for selecting surgery: serum Alb level (≥3.5 vs. <3.5) (odds ratio [OR], 42.487; 95% CI, 3.373 to 535.242; *p* = 0.004), PS (0, 1, 2, 3) (OR, 0.233; 95% CI, 0.067 to 0.811; *p* = 0.022) and clinical stage (OR, 0.278; 95% CI, 0.119 to 0.646; *p* = 0.003); details are shown in Table 2.

The results of decision tree analysis with three independent predictive factors from logistic multivariate analysis and the OS of each node are presented in Figure 3. Patients were divided according to serum Alb levels. Among patients with Alb ≥ 3.5 g/dL, 49 (77.8%) received surgery; among patients with Alb < 3.5 g/dL, 3 (23.1%) received surgery. Among patients with Alb ≥ 3.5 g/dL, most PS 0 patients (44, 88.8%) received surgery. Treatment outcomes were favorable in surgery patients with stage I, II (24, 100%) and those with stage III, IV (20, 76.9%). The 5-year OS rate was 93.3% and 55.7%, respectively. On the other hand, among patients with Alb < 3.5 g/dL, the 5-year OS rate, including 0% surgery patients with stage III, IV, was 0%. The 5-year OS of surgery patients with Alb < 3.5, stage I, II (3, 75.0%) was 33.3%, and those with Alb ≥ 3.5, PS > 0 (5, 38.5%) was 31.4% (Figure 3).

From these results, the recommended treatment strategy for patients of all ages with Alb ≥ 3.5 g/dL and PS 0 and stage I, II is curative surgery. On the other hand, the recommended treatment strategies for patients with Alb < 3.5 g/dL and stage III, IV patients are non-surgical palliative radiotherapy or best supportive care. Although other patients (those with Alb ≥ 3.5 g/dL, PS 0, and stage III, IV or Alb ≥ 3.5 g/dL and PS ≤ 1 or Alb < 3.5 g/dL and stage I, II) might select curative surgery, more detailed examination will be desired.

## 4. Discussion

The results of this study suggest that PS, stage classification, and Alb are factors influencing the selection of surgery in elderly patients with OSCC from decision tree analysis. The recommended treatment strategy for patients with Alb ≥ 3.5 g/dL and PS 0 and stage I, II of OS 93.3% is curative surgery. In contrast, non-surgical treatment is recommended for patients with Alb < 3.5 g/dL and stage III, IV of OS 0%. The PS is an indicator that can provide an easy, comprehensive assessment of general conditions based on the degree of limitation of daily activities. As PS worsens, surgery may be avoided due to the assumption of low surgical tolerance and a further decline in postoperative PS. Advanced cancer cases such as Stages III and IV are suspected to be more common in elderly patients in poor general conditions with more than PS 2, suggesting that disease detection is delayed as PS gets worse. This is expected to be related to social factors such as the elderly’s lack of awareness of their medical condition, economic reasons, and social factors such as the need for an attendant.

Decision tree analysis makes understanding mutual relationships among factors easy because the factors are automatically selected in order of their strong influence on the target variable and are arranged hierarchically. A decision tree is easy to understand visually and can be used to assist in deciding on a treatment plan [10].

Karnofsky Performance Status (KPS) < 90 was reported as the stand-alone negative independent prognostic factor for OS. They concluded that very elderly HANC patients (aged ≥ 80) could have favorable outcomes with curative therapy, and advanced chronological age alone should not prohibit patients from receiving treatment [4]. In our study, PS < 1 is the significant factor for selecting surgery and a good prognosis. The KPS < 90 is nearly equal to ECOG-PS < 1 and has the same result as our study. Sanabria et al. showed that noncurative treatment for resectable HANC may be favorable over curative intent treatment when patients had low PS, high clinical stage, or severe comorbidity. They reported that substandard treatment was offered to 19.9% of patients. Associated factors related to the selection of substandard treatment were higher age, oropharynx and hypopharynx tumor site, severe comorbidity, advanced clinical stage, and low KPS [11]. Wilson et al. reported that very elderly HNSCC patients with ≥80 years old could have favorable outcomes with curative therapy, and advanced chronological age alone should not prohibit patients from receiving treatment. PS, rather than age, appears to have more influence on treatment outcomes than age alone [4].

Nutritional status can assess immune function, wound healing, and risk of postoperative complications. Poor nutritional status has been associated with various factors, including morbidity, postoperative recovery, functional disability, length of hospital stay, and mortality. In addition, a preoperative nutritional assessment may be important in determining the course of treatment for elderly cancer patients, especially among the elderly and sick, since the physical and social effects of the disease frequently result in poor nutritional status [12]. Alb has a long half-life of approximately 21 days and is an indicator of mid to long-term nutritional status. It is also an indicator of cachexia since liver albumin production is inhibited by malignancy and inflammation. Therefore, a decrease in Alb level in patients with HANC may be due to either malnutrition associated with dysphagia or other problems, obstruction of the upper gastrointestinal tract system, or cancer progression [13]. Poor nutritional status in patients with HANC is associated with tumor progression and survival. Patients with low serum Alb levels (<3.5 g/dL) before treatment experienced approximately six-fold increases in the risks of tumor progression and cancer-specific and overall mortality compared to the findings in their counterparts [14]. In this study, patients with lower Alb levels (<3.5 g/dL) were selected significantly for the non-surgery group, and the OS was 33.3% in stages I, II, and 0% in stages III, IV. Therefore, Alb is an easy and significant factor in deciding a treatment plan for elderly patients.

Moreover, GNRI, based on Alb levels and the present and ideal body weight, is a simple screening tool to predict the risk of malnutrition and mortality in patients. Low GNRI (≤98) was associated with a poor prognosis in patients with OSCC than high GNRI in our previous study [6]. In the present study, the nutritional indices, including Alb, GNRI, and BMI, are related to selecting surgery and a good prognosis for elderly patients.

G8 screening is also considered effective in the functional assessment of elderly OSCC patients [15]. The G8 is calculated based on nutritional status, physical function, regular medications, mental status, and age and is one of the easiest screening tools to evaluate. Three of the eight questions are related to diet and nutritional status, indicating the importance this assessment method gives to the nutritional status of the patients. This opinion corresponds to the results obtained in this study. Yamada et al. reported that patients with G8 score less than 11.5 was difficult to treat, and the prognosis was poor [16]. The question data of the food intake decline and weight loss for 3 months were missed, but they correspond with BMI, GNRI, and Alb in our study. The data of more than three prescriptions and health status compared with same-age people were missed. Therefore, including G8 evaluation would be helpful in selecting elderly patients in future studies.

The limitations of this study include a single-institute design, which limits the number of cases and may have a bias in the selection of cases. In addition, since various patient assessment measures have been reported in recent years, more measures need to be evaluated. Further studies are desirable in the future, such as multi-institutional joint research, increasing the number of cases and evaluation measures.

In conclusion, according to the decision tree analysis, curative surgery is the recommended treatment strategy for patients of all ages with Alb ≥ 3.5 g/dL, PS 0, and stage I, II. On the other hand, the patients with Alb < 3.5 g/dL and stage III, IV patients are recommended non-surgical palliative radiotherapy or best supportive care. The 5-year OS rate among patients with Alb ≥ 3.5 g/dL, PS 0, and stage I, II was 93.3%, among patients with Alb < 3.5 g/dL and stage III, IV was 0%.

## Figures and Tables

**Figure 1 dentistry-11-00006-f001:**
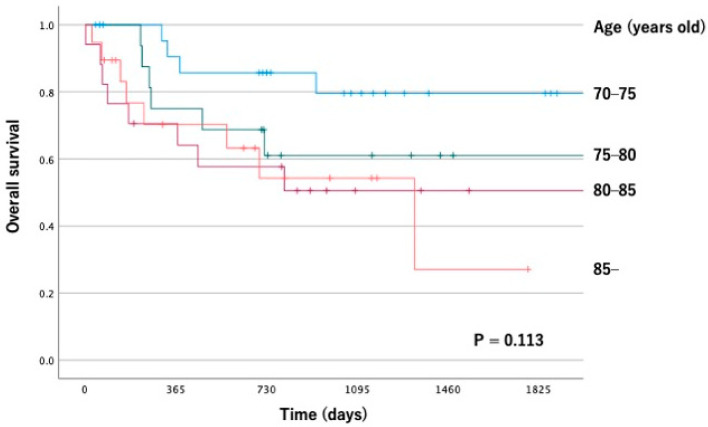
Kaplan–Meier survival curve according to the age group.

**Figure 2 dentistry-11-00006-f002:**
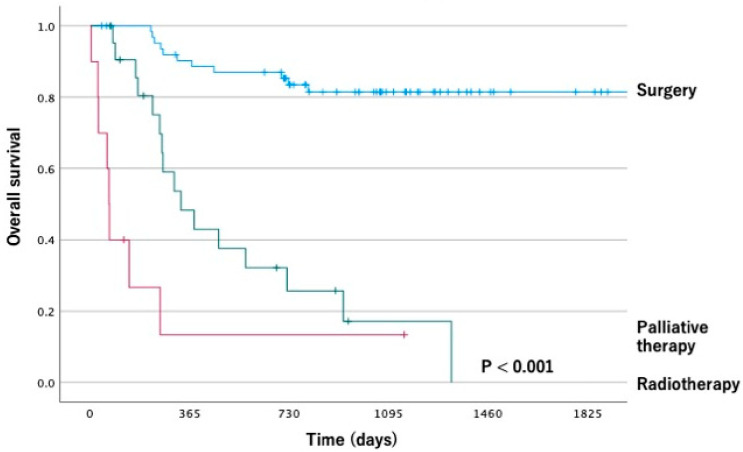
Kaplan–Meier survival curve according to the treatment group.

**Figure 3 dentistry-11-00006-f003:**
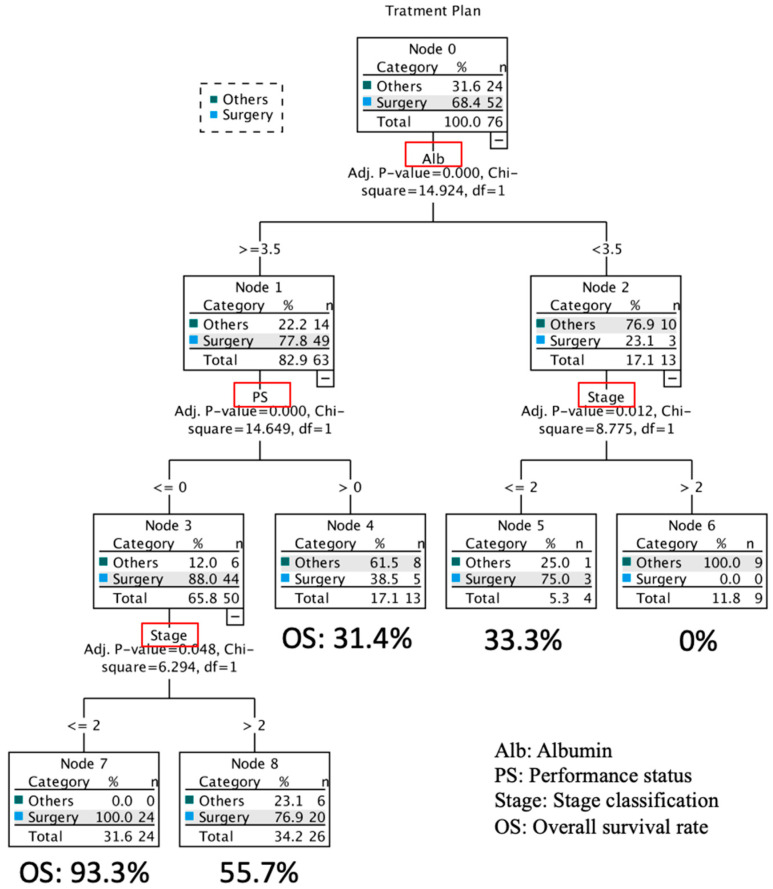
Decision tree analysis of treatment strategies. The items in red box were the three independent predictive factors from logistic multivariate analysis.

**Table 1 dentistry-11-00006-t001:** Clinical characteristics for surgery group and non-surgery group.

		Surgery (*n* = 52)	Non-Surgery (*n* = 24)	*p*
Sex	Male	34 (65.4%)	11 (45.8%)	0.107
	Female	18 (34.6%)	13 (54.2%)	
Age	70–74	19 (36.5%)	5 (20.8%)	0.089
	75–80	13 (25.0%)	3 (12.5%)	
	80–84	11 (21.2%)	6 (25.0%)	
	≥85	9 (17.3%)	10 (41.7%)	
PS	0	47 (90.4%)	14 (58.3%)	0.004 **
	1	4 (7.7%)	5 (20.8%)	
	2	0 (0%)	4 (16.7%)	
	3	1 (1.9%)	1 (4.2%)	
Presence illness	Presence	44 (84.6%)	21 (87.5%)	0.740
	Absence	8 (15.4%)	3 (12.5%)	
CCI	0	43 (82.7%)	14 (58.3%)	0.087
	1	3 (5.8%)	4 (16.7%)	
	2	5 (9.6%)	6 (25.0%)	
	3	1 (1.9%)	0 (0%)	
Primary site	Mandibular gingiva	21 (40.4%)	10 (41.7%)	0.221
	Tongue	17 (32.7%)	2 (8.3%)	
	Buccal mucosa	8 (15.4%)	5 (20.8%)	
	Maxillary gingival	4 (7.7%)	4 (16.7%)	
	Maxillary sinus	0 (0%)	1 (4.2%)	
	Floor of mouth	1 (1.9%)	1 (4.2%)	
	Others	1 (1.9%)	1 (4.2%)	
T classification	T1	14 (26.9%)	0 (0%)	0.002 **
	T2	18 (34.6%)	4 (16.7%)	
	T3	4 (7.7%)	3 (12.5%)	
	T4a	14 (26.9%)	11 (45.8%)	
	T4b	2 (3.8%)	6 (25.0%)	
N classification	N0	36 (69.2%)	10 (41.7%)	0.004 **
	N1	6 (11.5%)	3 (12.5%)	
	N2b	10 (19.2%)	6 (25.0%)	
	N2c	0 (0%)	5 (20.8%)	
Stage classification	I	14 (26.9%)	0 (0%)	<0.001 **
	II	16 (30.8%)	2 (8.3%)	
	III	2 (3.8%)	4 (16.7%)	
	IVA	18 (34.6%)	11 (45.8%)	
	IVB	2 (3.8%)	6 (25.0%)	
	IVC	0 (0%)	1 (4.2%)	
Serum albumin level (g/dL)	<3.5	3 (5.8%)	10 (41.7%)	<0.001 **
	≥3.5	49 (94.2%)	14 (58.3%)	
BMI (kg/m^2^)	<18.5	5 (9.6%)	3 (12.5%)	0.042 *
	18.5 to 25	28 (53.8%)	14 (58.3%)	
	≥25	19 (36.5%)	1 (4.2%)	
GNRI	<98	6 (11.5%)	7 (29.2%)	0.010 *
	≥98	46 (88.5%)	11 (45.8%)	

* *p* < 0.05, ** *p* < 0.01. Performance Status (PS), Charlson Comorbidity Index (CCI),Body Mass Index (BMI), Geriatric Nutritional Risk Index (GNRI).

**Table 2 dentistry-11-00006-t002:** Logistic multivariate analysis for selecting surgery in elderly patients with OSCC.

	B	SD	Wald	Exp	95% CI (min–max)	*p*
Alb (≥3.5 vs.<3.5)	3.749	1.293	8.413	42.487	3.373–535.242	0.004 **
PS (0,1,2,3)	−1.459	0.637	5.236	0.233	0.067–0.811	0.022 *
Stage	−1.281	0.431	8.839	0.278	0.119–0.646	0.003 **

* *p* < 0.05, ** *p* < 0.01.

## Data Availability

The data presented in this study are available on request from the corresponding author. The data are not publicly available due to ethical approvement.

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
