# Peer review of "Convenient Decision Criteria for Surgery in Elderly Patients with Oral Squamous Cell Carcinoma"

_dentistry, 2022, doi:10.3390/dj11010006_

Round 1

Reviewer 1 Report

Could provide supplemental document on how PS was determined. Additional factor like stress hormones in blood can be looked into as one of the indicators to determine treatment outcome. 

Reviewer 2 Report

This is an interesting research article regarding the criteria for selecting surgery or non in elderly patients with oral squamous cell carcinoma.

The overall writing is good. This paper is acceptable for publication provided that the authors proceed to some minor revisions listed below and writing improvements.

1. Surgery and not surgery groups are not distributed the same.

2. Why you perform only radiation therapy to the non surgery group and not also chemotherapy? Is there any specific protocol?

3. page 4 line 129: Can you please explain were have you been based and choose the normal range of serum Alb levels and the undernourished levels.

Reviewer 3 Report

The reviewer really appreciates the effort of the authors to accomplish this study.  The study is well designed and conducted nicely, however, the reviewer notice several major scopes of improving the manuscript to make it suitable for publication.

The Introductions part is very short and missing the logical background of the study. The author can improve this section by adding a review of some previous literature. The author can add a paragraph giving the problem statement and rationale of the current study.

The specific objectives of the study are missing in the manuscript.

The reviewer's understanding is “this was a retrospective study where the authors intend to figure out the decision-making criteria chosen by the surgeon/ surgical team member to deal with OSCC cases treated in the Department of Oral and Maxillofacial Surgery, the University of Tsukuba Hospital during 2015 to 2018. The author collected the patient-related information from the hospital database and performed relevant statistics to correlate the parameter chosen by the surgeon/ surgical team to decide on the surgical or non-surgical   approach to deal with the cases comparing with the success/ survival rate of the patient”

Currently, the methodology section expressed the view of the surgeon (which is usually used in case report studies) rather than the researcher's point of view which I think is more appropriate for such kind of study.     

The result of the study can be expressed either by the table/ figure/ chart or by the descriptive statement. Repetition of the same result both in descriptive text and table/ figure/ chart should be avoided. If the author prefers to use both, the descriptive text should be limited to stating the most important (significant) result obtained after statistical analysis that leads to accepting/rejecting the hypothesis/ null hypothesis used in the study.

Figure 3 graphics quality is poor. Please either use a word file or increase the DPI and pixel ratio of the image.   

In the discussion section, there is a repetition of statistics including the p-value of several results.

In conclusion, the survival/ success rate observed in the current study following the recommendation made by the author should be mentioned.  

Round 2

Reviewer 3 Report

The revised version has improved